# *Azospirillum brasilense* Bacteria Promotes Mn^2+^ Uptake in Maize with Benefits to Leaf Photosynthesis

**DOI:** 10.3390/microorganisms10071290

**Published:** 2022-06-25

**Authors:** Alexandra B. Housh, Spenser Waller, Stephanie Sopko, Avery Powell, Mary Benoit, Stacy L. Wilder, James Guthrie, Michael J. Schueller, Richard A. Ferrieri

**Affiliations:** 1Missouri Research Reactor Center, University of Missouri, Columbia, MO 65211, USA; alibauer1109@gmail.com (A.B.H.); sgwxhv@mail.missouri.edu (S.W.); s9207945@atsu.edu (S.S.); apgg4@mail.missouri.edu (A.P.); mvbenoit@mail.missouri.edu (M.B.); wildersl@missouri.edu (S.L.W.); guthriejm@missouri.edu (J.G.); schuellerm@missouri.edu (M.J.S.); 2Chemistry Department, University of Missouri, Columbia, MO 65211, USA; 3School of Natural Resources, University of Missouri, Columbia, MO 65211, USA; 4Biochemistry Department, University of Missouri, Columbia, MO 65211, USA; 5Division of Plant Sciences and Technology, University of Missouri, Columbia, MO 65211, USA; 6Interdisciplinary Plant Group, University of Missouri, Columbia, MO 65211, USA

**Keywords:** plant growth promoting bacteria, maize, manganese-52, carbon-11, chlorophyll

## Abstract

*Azospirillum brasilense* is a prolific grass-root colonizing bacteria well-known for its ability to promote plant growth in several cereal crops. Here we show that one of the mechanisms of action in boosting plant performance is through increased assimilation of the micronutrient manganese by the host. Using radioactive ^52^Mn^2+^ (t_½_ 5.59 d), we examined the uptake kinetics of this micronutrient in young maize plants, comparing the performance of three functional mutants of *A. brasilense*, including HM053, a high auxin-producing and high N_2_-fixing strain; *ipdC*, a strain with a reduced auxin biosynthesis capacity; and FP10, a strain deficient in N_2_-fixation that still produces auxin. HM053 had the greatest effect on host ^52^Mn^2+^ uptake, with a significant increase seen in shoot radioactivity relative to non-inoculated controls. LA-ICP-MS analysis of root sections revealed higher manganese distributions in the endodermis of HM053-inoculated plants and overall higher manganese concentrations in leaves. Finally, increased leaf manganese concentration stimulated photosynthesis as determined by measuring leaf fixation of radioactive ^11^CO_2_ with commensurate increases in chlorophyll concentration.

## 1. Introduction

Manganese (Mn) is an essential micronutrient for plant growth and development, and while it is required in only tiny amounts, it has been implicated as a cofactor in sustaining metabolic roles within different plant cell compartments [1,2]. Of note is its role in the photosynthesis process, where it is an essential cofactor for the oxygen-evolving complex of the photosynthetic machinery. Here, Mn sparks the photosynthesis process by splitting water after Photosystem II (PSII) fixes light to initiate the conversion of CO_2_ and water into complex carbohydrates [3,4]. It is called the element of life, for without the water-splitting reaction, the entire machinery fails to perform. At least 80% of all plant Mn is contained within the chloroplasts and is associated with PSII-enriched particles [5].

As with many of the essential micronutrients, there exists a fine range of Mn concentrations that are needed to sustain life processes such as photosynthesis, below which plants exhibit severe symptoms of deficiency, often dying as the end result, and above which severe toxicity symptoms result, which, too, can result in plant death. Prior studies addressing Mn deficiency in plants have largely shown that the photosynthetic process is affected by inhibition of electron donation at the PSII site, resulting in the cessation of electron transport activity [6,7,8,9]. As Mn becomes immobile in plants once deposited in leaf chloroplasts, deficiency symptoms first develop on the growing points and the plant leaves. The common symptoms are interveinal chlorosis, which manifests along the leaf edges. If Mn deficiency is not detected early and corrected, serious symptoms can occur in crops. Most noteworthy is a drop in photosynthesis and chlorophyll concentration [10]. Such a decrease will affect many processes of plant growth and development, including grain fill and quality, lodging (as Mn is involved in lignin biosynthesis), and compromised immunity response systems, making the plant vulnerable to pathogens and pests. 

The deleterious effect of Mn toxicity is often observed in the shoots as stunted growth, chlorosis, crinkled leaves, and brown lesions. However, the response of plants to excess Mn is often influenced by the age of the plant as well as the environmental and soil conditions [11]. While plant Mn management is an age-old problem, the physiological mechanisms of Mn toxicity and tolerance are still rather vague [12]. Several reports suggest a role for excess Mn in the induction of oxidative stress, and in fact, a very early study suggested that Mn-induced chlorosis was not caused by inhibition of chlorophyll synthesis but rather by photooxidation of chlorophyll [13]. Even so, there is also compelling evidence that suggests excess Mn can inhibit chlorophyll biosynthesis [14].

In agriculture, there is increasing interest in the utilization of plant-growth-promoting bacteria (PGPB) to improve both the yield and nutritional value of crops. These organisms can activate physiological and biochemical responses within their host for mutual benefit to build natural tolerances to environmental stresses and thereby reduce losses in the field [15,16,17,18,19,20,21]. Within the class of PGPB, *Azospirillum brasilense,* a Gram-negative bacterium, is perhaps the best-studied and has been commercialized as crop inoculants for maize and wheat [22,23,24,25]. Unlike rhizobia, which form an intracellular symbiosis with their legume hosts, PGPB do not induce the formation of observable plant structures or nodules. Further, they are also not usually major components of the soil microflora. However, they are prolific grass-root colonizers and, as epiphytes, are typically found growing on the root surface in very high numbers and with no observable host stress symptoms [26]. 

Enhancement of plant growth by *A. brasilense* and other PGPB has been attributed to several mechanisms of action, including N_2_-fixation [19], their ability to synthesize plant-relevant hormones such as auxin, impacting root growth traits [21], and their ability to promote host uptake of essential micronutrients such as iron [21] and zinc [27]. Here we wanted to examine *A. brasilense*’s influence on host Mn^2+^ uptake, given the importance of this micronutrient in supporting the plant’s photosynthetic machinery.

## 2. Materials and Methods

### 2.1. Bacteria Growth and Root Inoculation

Functional mutants of *A. brasilense* were obtained via a material transfer agreement between the corresponding author’s institution and the Federal University of Paraná (UFPR, Curitiba, PR CEP 81531-980, Brazil). HM053 is a mutant strain that is constituently expressed in the *Nif* gene, resulting in hyper N_2_-fixing capacity, as characterized in prior screening studies [28,29], but also possesses a high auxin (indole-3-acetic acid)-producing capacity [21]. The FP10 mutant was characterized in the original work of Pedrosa and Yates [30] and is deficient in N_2_-fixation capacity but still produces auxin, albeit at a lesser rate than HM053 [21]. The *ipdC* mutant reflects a disruption in the indole-3-pyruvate decarboxylase gene (ipdC) that codes a key enzyme of the indole-3-pyruvic acid pathway in auxin biosynthesis [31]. The resultant knock-out strain exhibits a significant reduction in auxin biosynthesis to a level of 10% of that of the wild-type strain [32]. 

Functional mutants were grown in a liquid NFbHP-lactate medium following published procedures [19]. The bacterial cultures were grown in 25 mL flasks in a shaking incubator set to 30 °C and 130 rpm until OD_600_ = 1.0 (optical density at 600 nm, corresponding to 10^8^ cells mL^−1^) was reached. Cultures were then washed with sterile water and diluted to approximately 10^6^ to 10^8^ colony forming units (CFU) per mL volume. Root inoculation involved adding 1 mL of inoculum to a Petri dish containing 10 maize seedlings (2 days after germination) and rocking in the shaking incubator for two hours. Seeds were placed into plastic seed germination pouches (PhytoAB, Inc., San Jose, CA, USA) for five days before transplanting to hydroponics.

### 2.2. Plant Growth

Maize kernels (Hybrid 8100, Elk Mound Seed Co., Elk Mound, WI, USA) were surface sterilized, germinated in the dark and then transferred to hydroponics for continued growth with Hoagland’s nutrient added to the hydroponics solution every 5 days to replenish depleted nutrients. Growth conditions consisted of 12-h photoperiods, 500 μmol m^−2^ s^−1^ light intensity, and temperatures of 25 °C/20 °C (light/dark) with humidity at 70–80% controlled by a commercial growth chamber (Model PGC-15, Percival Scientific, Perry, IA, USA). Plants were studied after 3 weeks of growth.

### 2.3. Plant ^52^Mn^2+^ Uptake Studies

Radioactive ^52^Mn has a half-life of 5.59 d and decays 29.6% by positron emission. Positron annihilation resulting in two coincident gamma rays at 511 keV energy makes this radioisotope ideal for tracking plant uptake of manganese via gamma counting. For these studies, we purchased a 37 MBq dose of ^52^Mn, which was produced on the University of Wisconsin PETtrace cyclotron (General Electric Healthcare, Waukesha, WI, USA) by ^nat^Cr(p,n)^52^Mn using 16 MeV protons. The procedures for radioisotope separation are detailed in the following sentences [33]. Because the radioisotope was produced under no-carrier-added conditions using a ^nat^Cr target, the entire 37 MBq shipment had only 5 μg of stable ^55^Mn. Following irradiation, targets were etched with HCl, and the resulting solution was diluted with ethanol. This was done to reach a condition where dissolved Mn^2+^ anionic chloride complexes extract onto strong anion-exchange resin while hydrated Cr^3+^ passes through. Upon switching to a fully aqueous 6 M HCl solution, ^52^Mn^2+^ was released from the resin bed. The entire procedure of trap-and-release, alternating from ethanolic to aqueous mixtures, was repeated three times on three separate small AG-1 × 8 columns. The purified dose of ^52^Mn^2+^ was recovered in 0.5 mL of dilute aqueous HCl for shipment.

Upon receipt of the shipment, the dose was pH neutralized and diluted further with deionized water for administration. Study plants were removed from their hydroponics solution, washed in deionized water, and placed in a glass beaker filled with 200 mL of deionized water. The radioisotope at a 0.74 MBq dose was syringed into this volume for dynamic uptake studies (Figure 1). Carrier manganese was not introduced into the experiment. Hence, the 0.74 MBq dose of ^52^Mn had approximately 100 ng of ^55^Mn carrier. After ^52^Mn^2+^ was administered to the root solution, the plant was monitored for 3 h before separating shoots and roots for tissue counting of radioactivity. During that time, levels of radioactivity were monitored using a radiation detector (Eckler & Ziegler, Inc., Berlin, Germany 1-inch Na-PMT, photomultiplier tube gamma radiation detector) affixed to the plant 4 cm above the base of the stem (Figure 1), which provided dynamic feedback on radioisotope uptake. Data were acquired at a 1 Hz sampling rate using a 0–1 V analog input into an acquisition box (SRI Inc., Torrance, CA, USA). 

After 3 h of dynamic measurements, the plants were removed from the solution, washed in deionized water 3 times and separated from the shoots. Tissues were then counted in a NaI gamma counter to measure the amount of ^52^Mn radioactivity they possessed, leveraging the 511 keV gamma-ray signature the isotope generates from positron annihilation. The combined tissue activities were decay corrected to a common timepoint and called “*biological assimilation*” since it was not possible to distinguish ^52^Mn assimilation by the microorganisms growing on the root surface from that taken up by the host plant. This value reflected the amount of radioactivity that was removed from the solution in which the roots were submerged. Values of “*shoot allocation*” were then calculated by dividing shoot radioactivity by the biological assimilation value.

### 2.4. Production and Administration of Radioactive ^11^CO_2_

^11^CO_2_ (t_½_ 20.4 min) was produced on the GE PETtrace Cyclotron located at the Missouri Research Reactor Center following prior published procedures utilizing the ^14^N(p,α)^11^C nuclear transformation [34,35,36]. The ^11^CO_2_ was collected and administered to live plants in a flow of air at 22 °C and under light conditions of 500 μmol m^−2^ s^−1^. A PIN diode radiation detector (Carroll Ramsey Associates, Berkeley, CA, USA) was used to provide continuous monitoring of radiation levels, indicating the amount of ^11^CO_2_ fixed by the plant.

### 2.5. Inductively Coupled Plasma-Mass Spectrometry (ICP-MS) of Root and Leaf Tissue for Mn^2+^

Details on ICP-MS analyses of leaf and root tissues for full elemental analysis can be found in our prior published work [21]. In brief, central portions of leaf 2 (excluding the tip and base of the leaf), as well as roots, were harvested, dried, and digested in nitric acid using a Milestone Ethos Plus (Milestone SRL, Sorisole, Italy) microwave digestion system. Samples were then analyzed on a PerkinElmer NexION ICP-MS and matched against NIST standards. Each biological replicate reflected our binning dried tissues from three separate plants for a total of N = 6 replicates.

For Laser Ablation-ICP-MS, roots were removed from shoots, and a 3-cm section of non-lateral root tissue was further excised above the apical meristem. Roots were sectioned in OCT embedding media to 100 μm thickness (Fisher Scientific Inc., Hampton, NH, USA) and placed on quartz microscope slides for freeze-drying in a FreezeZone 1 dryer (Labconco Corp., Kansas City, MO, USA) before ablation. Sections were ablated using an Analyte.193 ultra-short pulse excimer laser ablation system (Photon-Machines Inc., Redmond, WA) with 50% laser output, 40 µm spot size, 10 Hz rep rate, 10 µm/s scan speed, linear trace, and 0.9 L/min He sweep gas flow. The laser ablation system was coupled to a PerkinElmer NexION 300X (PerkinElmer, Shelton, CT, USA) operated in kinetic energy discrimination mode at 3.5 mL min^−1^ helium cell gas flow, 1.1 L min^−1^ argon nebulizer gas flow with a 50 ms dwell time on the laser pulse. Each biological replicate reflected our analyzing one root section from separate plants for N = 6 replicates of non-inoculated control and HM053-inoculated plants.

### 2.6. Leaf Chlorophyll Analysis

Leaf chlorophyll concentrations were measured using our prior published procedures [37]. In short, leaf 2 samples were ground in liquid nitrogen and extracted in acetone. Components of the extract were resolved by the thin-layer chromatography method. Chlorophyll a & b components were removed and resuspended in acetone and their UV absorbances were measured at 663 nm and 645 nm, respectively.

## 3. Results

Data from the dynamic root-shoot transport of radioactive ^52^Mn is shown in Figure 2A–D for non-inoculated control plants and plants inoculated with HM053, *ipdC* and FP10 strains of *A. brasilense*, respectively. Results here show that HM053 can stimulate significant increases in ^52^Mn transport relative to non-inoculated control plants, which did not show a change in stem radioactivity over the 3-h window of incubation with tracer. In contrast, *ipdC* did not stimulate ^52^Mn transport. However, FP10 showed signs of increased tracer transport though to a lesser extent than HM053.

Data from tissue counts are presented in Figure 3A,B. Since it was not possible to distinguish ^52^Mn^2+^ assimilation by the root-associating bacteria from that of their host plant, Figure 3A reflects the biological assimilation of radiotracer over 3 h of the combined actions of bacterial and host plant assimilation. Inoculation with HM053 and FP10 bacteria significantly increased biological assimilation relative to control. Inoculation with *ipdC* did not affect the amount of ^52^Mn^2+^ assimilated relative to control. Furthermore, shoot activity levels for HM053-inoculated plants were significantly elevated over those of control plants. Inoculation with *ipdC* showed no significant change in shoot activity relative to controls, which correlated well with the dynamic data in Figure 2C. Finally, shoot activity from FP10-inoculated plants showed a slight elevation relative to controls, but the data was not significant. Here, too, the slight increase in static tissue counts is reflected by the dynamic data presented in Figure 2D, showing a slight increase in stem transport of Mn^2+^.

Analysis of plant tissues by ICP-MS provided additional validation that bacterial inoculation increased host tissue Mn levels. Results in Figure 4 reflect root (Figure 4A) and leaf (Figure 4B) Mn concentrations as measured by the natural abundance ^55^Mn signature. Inoculation with HM053 resulted in significantly lower Mn concentrations in roots but significantly higher Mn levels in shoots compared with non-inoculated control plants.

These results are related to the fact that HM053-inoculated plants exhibited higher stem transport of the ^52^Mn over the short 3 h window. We expect, therefore, that over the longer 3-week time period of plant growth, a higher leaf Mn level would manifest as root Mn is continuously transported aboveground. Contrary to this, *ipdC*-inoculated plants showed higher root Mn levels than control plants and significantly less leaf Mn owing to the inability of Mn to transport efficiently aboveground. Finally, FP10 inoculation caused similar results as HM053-inoculated plants with higher leaf Mn observed.

To gain further insight into the effects of HM053 bacteria on Mn transporter activity at cellular scales within the plant root cells, we conducted a subset of laser ablation ICP-MS measurements on 100 μm thick freeze-dried root sections (Figure 5). Here, comparisons were made in the relative Mn-55 ion signal across the epidermis, cortical cells, endodermis and core stele surrounding the vasculature of non-inoculated control plants (Figure 5A) and HM053-inoculated plants (Figure 5B) with average values graphically presented in Figure 5C. Manganese levels within the endodermis were significantly elevated in HM053-inoculated plants as compared to controls.

Finally, we examined the influences of microbial inoculation with the three functional mutants of *A. brasilense* on the host plant’s photosynthetic machinery. In Figure 6, the results of leaf ^11^CO_2_ fixation are presented (Figure 6A), along with the results of our analysis of leaf chlorophyll a (Figure 6B) and chlorophyll b (Figure 6C). HM053 inoculation significantly increased ^11^CO_2_ fixation as well as increased levels of chlorophyll a and b concentration proportionately relative to non-inoculated control plants, resulting in an unchanged R_a/b_ value (see Figure 6C where R_a/b_ values were documented). These results may be related to the increased leaf Mn, which would up-regulate the photosynthetic machinery. Contrary to this, inoculation with *ipdC* reduced fixation, as well as reduced chlorophyll a, but not chlorophyll b, resulting in a lower R_a/b_ value. These results have some relationship with the lowered leaf Mn. FP10 inoculation had no effect on fixation nor on chlorophyll biosynthesis relative to non-inoculated control plants. We suspect the effects of this strain on Mn transport were not sufficiently elevated to impact the photosynthetic machinery.

## 4. Discussion

Mn homeostasis-maintaining compartmented metabolic processes such as photosynthesis, glycosylation and ROS scavenging are thought to be mediated by several transport proteins from diverse gene families. In the past, Mn transporters have been identified at the molecular level in many eukaryotic organisms [38], and while past work on Mn homeostasis in higher plants has mainly focused on Arabidopsis and rice [39,40], it is only recently that transporters involved in the uptake and subcellular distribution of Mn have been characterized in a wider range of plant species [2], including the zinc-regulated transporter (ZRT), iron-regulated transporter (IRT)-like protein (ZIP), natural resistance-associated macrophage protein (NRAMP), and yellow stripe-like protein (YSL) [2,41,42,43]. Active cycling of root transport proteins such as these may occur across the outer plasma membrane and the endomembrane compartments of the individual root cells and, along with their actions [44], appear to control metal transport activity by maintaining a polar distribution of proteins both at the outer root surface, where they serve to import metals from the soil matrix into the cytoplasm, and at the endodermis, where they serve to export metals across the hydrophobic Casparian strip into stele and vascular core, where, once in the xylem, these metals are free to move aboveground in the xylem driven by the plant’s transpiration stream [45].

The present work demonstrates that HM053 can promote uptake and whole-plant transport of Mn^2+^ under deficient conditions. As noted, there was a significant increase in both biological assimilation and stem transport using radioactive ^52^Mn^2+^ as a tracer. To a lesser extent, we observed similar behavior with the FP10 strain. Once assimilated at the root surface, Mn^2+^ is presumed to travel through the apoplastic space between the root cells to reach the endodermis. However, symplastic cell-to-cell transport cannot be ruled out as ions move toward the inner endodermal ring. Once there, the endodermal ring can be a site rich in ion transporters, facilitating the trafficking of metal ions to the vascular core containing the xylem [46]. The endodermis also manifests as a natural barrier to ion trafficking in the form of the Casparian strip and suberin lamellae. Here, a layer of waterproof lignin and suberin polymer exists, which can force ion nutrients to pass into the symplast [47]. The Casparian strip has been linked with regulating the transport of metals, including Fe^3+/2+^, Zn^2+^ and Mn^2+^ [48]. Once in the vascular core, metal transport in the xylem should be dependent on water transport to aerial portions driven by leaf transpiration.

As validation of microbial influence on host Mn^2+^ ion transport, we observed significantly higher accumulations of natural abundance ^55^Mn within the root endodermis of HM053-inoculated plants than controls when LA-ICP-MS was applied to a subset of root sections. The overall effect was Mn accumulated significantly over time in the leaves of HM053-inoculated plants. This phenomenon is the same as what we reported in prior studies examining the effects of HM053 on iron uptake and its transport in maize [21], suggesting that there could be a direct mode of action by these bacteria affecting either the abundance of transport proteins or their level of activity in transporting metal ions. On the other hand, root-colonizing bacteria may influence host root cell morphology, perhaps increasing the apoplastic space between cells and impacting metal ion diffusional properties. We note that in our prior studies using transmission electron microscopy, HM053 caused a thickening of the Casparian bands [21], suggesting that these bacteria can influence root morphology at cellular scales.

The ability of HM053 to outperform the *ipdC* and FP10 strains of *A. brasilense* in promoting host Mn uptake and transport is possibly due to its high auxin-producing capacity. In our prior work [21], we noted that the microbial auxin-producing capacity of these *A. brasilense* mutant strains followed this order: HM053 > FP10 > *ipdC*.

Since tryptophan is a key precursor in auxin biosynthesis and derives from the shikimate pathway, there could be direct links between microbial Mn demands and their auxin-producing capacity. The shikimate pathway is composed of seven enzymatic reaction steps initiated by the coupling of phosphoenolpyruvate with D-erythrose 4-phosphate to produce chorismatic. Five of the seven enzymatic steps within this pathway rely on Mn as a cofactor. Hence, a high microbial auxin-producing capacity would lend itself to a higher turnover of the shikimate pathway with commensurate higher demands for Mn. We suspect this action could serve to increase Mn trafficking to the host plant, as evidenced in the present work.

Lastly, we note that with increased host Mn uptake and transport to foliar tissues, there is a clear improvement to the host plant’s photosynthetic machinery, as evidenced by the increased levels of ^11^CO_2_ fixation and leaf chlorophyll concentration. Similar behavior was seen in our prior outdoor trials carried out over the 2018 and 2019 growing seasons. Here, we showed that maize inoculated with HM053 exhibited 12% larger stem diameters and 23% increased leaf thickness with 58% higher chlorophyll concentration relative to non-inoculated control plants. These improved growth traits lent themselves to the overall effect of significantly increasing crop yields [49]. We suspect that the increase in foliar chlorophyll concentration with HM053 inoculation is a consequence of its ability to promote host uptake and transport of Mn to leaf tissues, thereby increasing the water-splitting reaction in the oxygen-evolving complex of PSII. However, it is beyond the scope of the present work to provide any more insight into the mechanisms of action here than was already discussed. For example, an alternate hypothesis to this scenario is that microbial inoculation causes an up-regulation of chlorophyll biosynthesis, which, in turn, promotes the photosynthetic machinery, thereby increasing the plant’s demand for Mn. To test this theory, we would need to remove the bacteria as a variable and examine the direct influence of Mn on chlorophyll biosynthesis. By directly infusing leaves with Mn in combination with administering ^11^CO_2,_ we should be able to examine how the flux of ‘new’ carbon (as ^11^C) into chlorophyll changes with the application of exogenous Mn.

In summary, the present work provides evidence to suggest that auxin-producing bacteria can promote plant growth by improving host Mn assimilation, which, in turn, can improve the plant’s photosynthetic capacity.

## Figures and Tables

**Figure 1 microorganisms-10-01290-f001:**
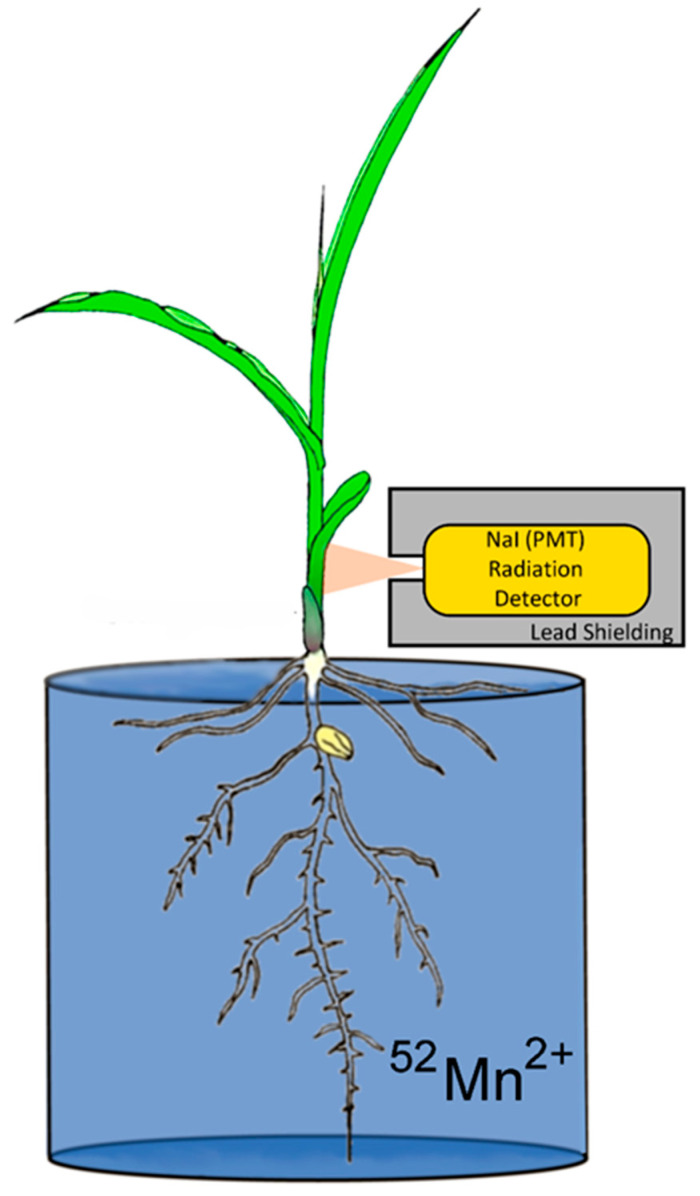
Experimental setup for monitoring plant uptake of radioactive ^52^Mn^2+^.

**Figure 2 microorganisms-10-01290-f002:**
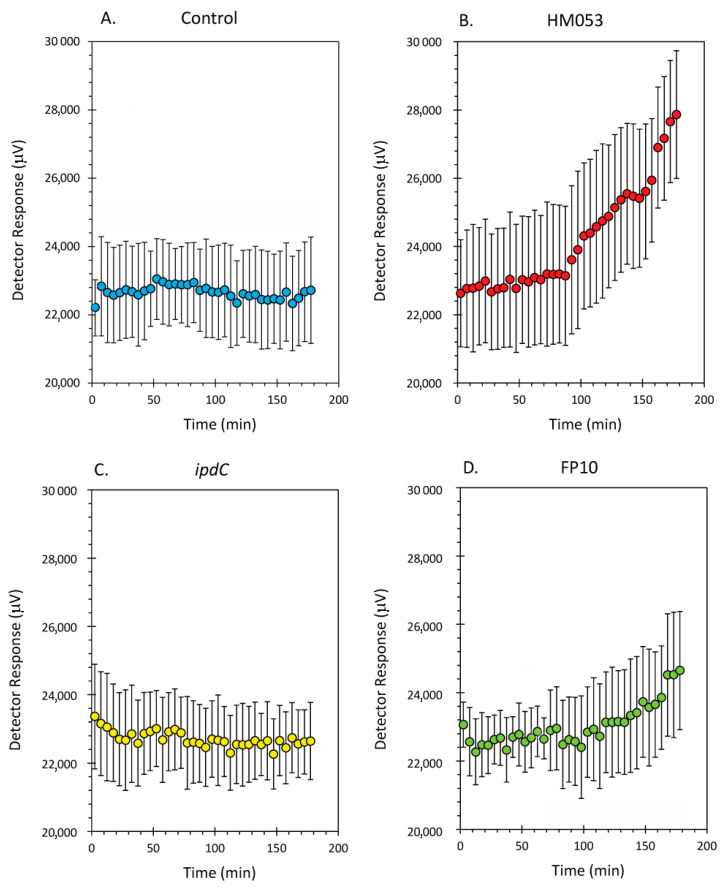
Dynamic time-activity curves showing transport of radioactive ^52^Mn in live maize plants. Using a fixed NaI gamma detector (as shown in Figure 1), the level of radioactivity transported through the plant stem was continuously monitored over 3 h. Panel (**A)**: time-activity data from non-inoculated control plants; Panel (**B)**: time-activity data from HM053 inoculated plants; Panel (**C)**: time-activity data from *ipdC* inoculated plant; Panel (**D)**: time-activity data from FP10 inoculated plants. Data was collected at a 1 Hz rate and was binned into 5 min intervals reflected as individual data points in these graphs. Error bars represent standard errors. The number of biological replicates was defined by the number of plants we examined per treatment (control, N = 6 plants; HM053-inoculated, N = 6 plants; *ipdC*-inoculated, N = 4 plants; FP10, N = 6 plants).

**Figure 3 microorganisms-10-01290-f003:**
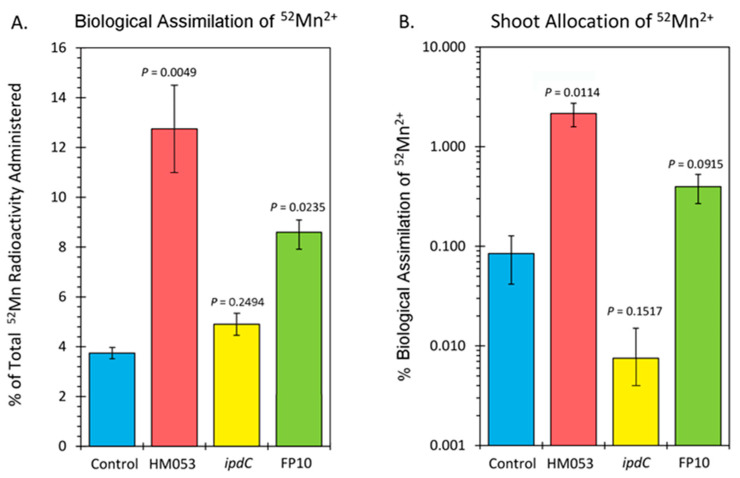
Summary of ^52^Mn^2+^ uptake studies. Panel (**A**): biological assimilation of ^52^Mn^2+^ radiotracer over 3 h representing the combined actions of microbial and host plant assimilation. Panel (**B**): allocation of assimilated ^52^Mn^2+^ to the shoots of the host plant. Data bars reflect mean values ± SE. The number of biological replicates was defined by the number of plants we examined per treatment (control, N = 6 plants; HM053-inoculated, N = 6 plants; *ipdC*-inoculated, N = 4 plants; FP10, N = 6 plants). *p* values < 0.05 were considered statistically significant.

**Figure 4 microorganisms-10-01290-f004:**
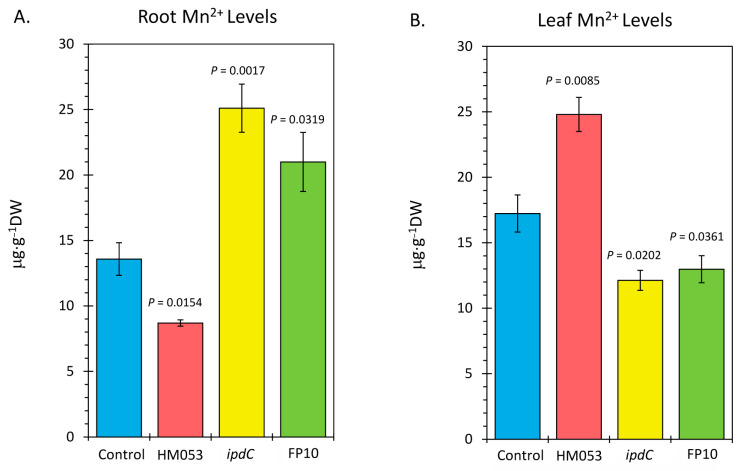
Inductively coupled mass spectrometry (ICP-MS) analysis of plant tissues for Mn-55. Panel (**A**): root concentrations presented in micrograms of manganese per gram of dry weight tissue (μg⋅g^−1^ DW). Panel (**B**): leaf concentrations presented in μg⋅g^−1^ DW. Data bars reflect mean values ± SE. The number of biological replicates is defined by the number of plants we examined per treatment (control, N = 6 plants; HM053-inoculated, N = 6 plants; *ipdC*-inoculated, N = 4 plants; FP10, N = 6 plants). *p* values < 0.05 were considered statistically significant.

**Figure 5 microorganisms-10-01290-f005:**
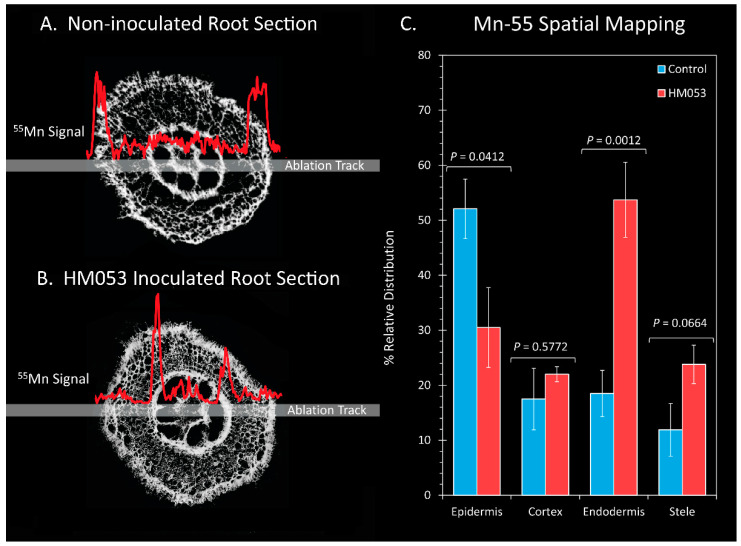
Laser ablation ICP-MS analysis reveals effect of HM053 inoculation on the spatial distribution of Mn-55 across root cells. Panel (**A**): The Mn-55 ion signal across a root section of a non-inoculated control plant. Panel (**B**): The Mn-55 ion signal across a root section of an HM053-inoculated plant. Panel (**C**): Average values ± SE on N = 6 replicates where each replicate was a root section taken from different plants. *p* values < 0.05 were considered statistically significant.

**Figure 6 microorganisms-10-01290-f006:**
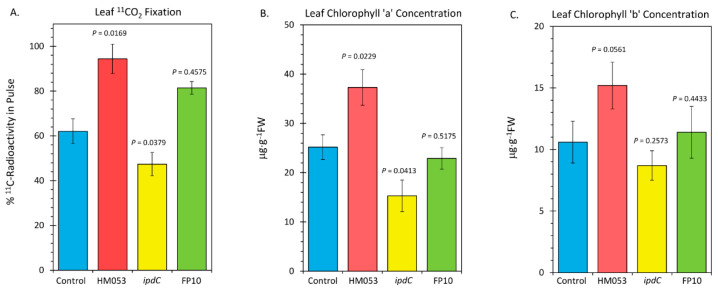
Metrics relative to performance of leaf photosynthesis change upon root inoculation with mutant strains of *A. brasilense* relative to non-inoculated control. Panel (**A**): Fixation of ^11^CO_2_ presented as percent of ^11^C-radioactivity in the pulse. Data were normalized to uniform leaf masses affixed within the leaf cuvette. Panel (**B**): Leaf concentration of chlorophyll a presented in micrograms per gram dry weight of tissue (μg⋅g^−1^ DW). Panel (**C**): Leaf concentration of chlorophyll b (μg⋅g^−1^ DW). Data bars reflect means ± SE. The number of biological replicates was defined by the number of plants we examined per treatment (Control, N = 6 plants; HM053-inoculated, N = 6 plants; *ipdC*-inoculated, N = 4 plants; FP10, N = 6 plants). Statistical significance was based on *p* < 0.05. Numeric ratios of chlorophyll a to chlorophyll b are also posted in Panel C as R_a/b_ values.

## Data Availability

Data will be provided on request to corresponding author.

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
