# Peer review of "Azospirillum brasilense Bacteria Promotes Mn2+ Uptake in Maize with Benefits to Leaf Photosynthesis"

_microorganisms, 2022, doi:10.3390/microorganisms10071290_

Round 1
Reviewer 1 Report
This manuscript describes how the inoculation of Azospirillum brasilense functional mutants HM053, ipdC, and FP10 affect manganase uptake in host maize plants. By administering the short-lived radioisotope Mn-52 to roots and monitoring aboveground radioactivity, the authors demonstrated that more manganese is transported aboveground in HM053 inoculated plants. In addition, based on the quantitative values of manganese concentration using ICP-MS and the results of tissue distribution of manganese using laser ablation ICP-MS, the authors claimed that HM053 inoculation facilitates manganese transport to the aboveground of maize plants. Finally, based on experiments with short-lived 11CO2 and the results of chlorophyll content quantification, the authors showed that increased leaf manganese content stimulates leaf photosynthetic activity.
The authors have published papers regarding the effect of PGPB on iron[21] and zinc[27] transport using the same experimental system, and this manuscript is the manganese article in the series of papers. This study aims to provide a comprehensive picture of the effects of PGPB on mineral transport and is therefore of great academic value.
The comments to make this manuscript more understandable are as follows.
1) Concerning Figure 3, the method for quantifying Mn-52 is missing. The method (gamma well- type detector?) should be provided in the section "2.3. Plant 52Mn2+ Uptake Studies".
2) In L126, it should be stated whether or not non-radioactive manganase was added with Mn-52.
3) In Figure 5AB, blue letters (55Mn Signal) and signal lines are hard to see. Please change the letters to white and the lines to red.
Author Response
Reviewer 1 (Author responses to reviewer comments are in italics)
1) Concerning Figure 3, the method for quantifying Mn-52 is missing. The method (gamma well- type detector?) should be provided in the section "2.3. Plant 52Mn2+ Uptake Studies". Thank you for picking up on this. I have added additional text (Lines 140-149) describing our methods for counting and quantifying radioactivity in plant tissues.
2) In L126, it should be stated whether or not non-radioactive manganese was added with Mn-52. This was a good point. The production of Mn-52 is no-carrier-added but the producers indicate that there is a tiny amount of exogenous Mn-55 mixed with the radioisotope. We estimated that per 0.74 MBq dose of Mn-52 administered to a live plant there was 100 ng of Mn-55. We did not add exogenous Mn-55 when tracer was introduced to the plant. Please see lines 116-118 and lines 130-131 for detailed explanation.
3) In Figure 5AB, blue letters (55Mn Signal) and signal lines are hard to see. Please change the letters to white and the lines to red. I agree. The labels are now white and the Mn-55 signal lines are now red.
Reviewer 2 Report
The study was well done, the description of the results and discussion is good. It is a valid approach to use radioactive Mn2+ and C11, but I miss further evidence on the transport mechanisms of Mn2+and the role of the bacteria in transport. Figure 5 is very interesting, but I cannot follow how the very basic model presented in Figure 7 is derived from it. How does bacterial inoculation alter the root cells to increase Mn2+ uptake? Only based on transporter abundance and activity? Based on the references ZRT, ZIP, NRAMP have been characterized in maize. Hence, it would be a simple add-on experiment to check if their expression is altered after inoculation. Why is Mn2+ first taken up by the epidermal cells and then transported apoplastically? Where are the bacteria located? Do they only live on the root surface or are they endophytes? Based on the observation of the root sections - have the authors observed any changes in the root structure? It would be great if the inoculated and non-inoculated root sections could be shown in the same size/using the same magnification to get a better idea of putative morphological changes.
Line 248 - brasilense
Author Response
Reviewer 2 (Author responses to reviewer comments are in italics)
The study was well done, the description of the results and discussion is good. It is a valid approach to use radioactive Mn2+ and C11, but I miss further evidence on the transport mechanisms of Mn2+and the role of the bacteria in transport. Figure 5 is very interesting, but I cannot follow how the very basic model presented in Figure 7 is derived from it. How does bacterial inoculation alter the root cells to increase Mn2+ uptake? Only based on transporter abundance and activity? The fact that HM053 resulted in more Mn accumulation in the inner endodermal ring suggests that bacteria are influencing either transporter abundance/activity, or cell morphology in the cortical region which could affect the apoplastic space between cells facilitating radial metal ion diffusion. Additional text was added to the Discussion reflecting these theories. Based on the references ZRT, ZIP, NRAMP have been characterized in maize. Hence, it would be a simple add-on experiment to check if their expression is altered after inoculation. This is a great suggestion but is not something we do in my plant radiotracer laboratory. Future collaborative studies will be established to enable us to examine this aspect. Additionally, electron microscopy of the outer cortical region of root sections could shed light on whether colonization of the host root surface influences root cell morphology. I do present evidence in the Discussion from our prior studies [see reference 21) where HM053 caused a thickening of the Casparian bands. More detailed electron microscopy studies of the cortical regions would have to be something for future studies.
Why is Mn2+ first taken up by the epidermal cells and then transported apoplastically? The general consensus in the science community is metals most often transport by this mechanism until the reach the endodermis. That is not to say symplastic cellular transport is not happening. The model I present in Fig. 7 reflects what most researchers consider to be the major mode of metal ion transport. I feel it is important to convey this visually in the paper so readers not familiar with the concepts of radial ion transport across the root can grasp the importance. I also feel it helps readers understand the importance of our experimental results presented in Fig. 5.
Where are the bacteria located? Do they only live on the root surface or are they endophytes? Please see line 72 where it is stated that A. brasilense is an epiphyte. There I cite our prior work [ref 21] using confocal microscopy showing root colonization. Based on the observation of the root sections - have the authors observed any changes in the root structure? It would be great if the inoculated and non-inoculated root sections could be shown in the same size/using the same magnification to get a better idea of putative morphological changes. I included a better digital photo of a root section for Fig. 5. Note that these images were acquired by a simple microscope/web cam on the LA-ICP-MS instrument and are not of sufficient quality for us to examine changes in cell morphology. As mentioned above more detailed studies using electron microscopy would be needed to answer some of the reviewer questions here.
Line 248 – brasilense Spelling was corrected.
Reviewer 3 Report
The topic is original and relevant showing that a PGPB species associated with maize contributes to the Mn uptake and mobilization in the plant, from roots to leaves, increasing the photosynthetic capacity of the plant and crop yield.
Overall, manuscript has a good quality. However, authors could improve the Keywords avoiding the repetition of words already included in the title.
Throughout the text, authors should replace "content" by "concentration".
Introduction:
-L59/60: move above.
M&M:
The experimental layout should be clearly described. Number of replicates is not clear: 4-6=? Precise number of replicates should also be included in Figures 2-4. Number of plants to be sampled for each determination should be indicated.
-L87: add "N2-fixation capacity"
Results:
Authors refer "correlation" but no correlation or regression data are presented. If authors have not determine the correlation coefficients, please replace the word "correlate" by "is apparently related with" or "has some relationship with".
Chlorophyll should be identified with lowercases "a" and "b", in the text and Fig. 6.
-L173: add "(Fig. 2)"
Discussion:
-L273: replace "deplete" by "defficient"
-L279: "Casparian strip": is Mn blocked by this band?
Author Response
Reviewer 3 (Author responses to reviewer comments are in italics)
The topic is original and relevant showing that a PGPB species associated with maize contributes to the Mn uptake and mobilization in the plant, from roots to leaves, increasing the photosynthetic capacity of the plant and crop yield. Overall, manuscript has a good quality. However, authors could improve the Keywords avoiding the repetition of words already included in the title. Keywords were changed to minimize repetition with the title.
Throughout the text, authors should replace "content" by "concentration". Corrections were made throughout the manuscript (see lines 24, 26,49, 341,344 and 347 where changes were made in the text. Also please see Fig. 6 caption where changes were made).
Introduction: L59/60: move above. This comment was vague. We did not move the line up in the Introduction as we felt it was best stated at that location in the paragraph.
Materials and Methods: The experimental layout should be clearly described. Number of replicates is not clear: 4-6=? Precise number of replicates should also be included in Figures 2-4. Number of plants to be sampled for each determination should be indicated. Changes were made for greater clarity. Please see lines 164 and following,167-168, 179-181 and figure captions.
-L87: add "N2-fixation capacity" Corrected – see lines 84 & 87
Results:
Authors refer "correlation" but no correlation or regression data are presented. If authors have not determine the correlation coefficients, please replace the word "correlate" by "is apparently related with" or "has some relationship with". Corrections were made – see lines 206-207, 237, 264-265, 267-268,344
Chlorophyll should be identified with lowercases "a" and "b", in the text and Fig. 6. Corrections were made to Fig. 6 as well as in the text of the manuscript
-L173: add "(Fig. 2)" I was not sure why Fig. 2 should be added to L173. It did not seem appropriate, so no edits were made.
Discussion:
-L273: replace "deplete" by "defficient" I replaced ‘deplete” with “deficient” in line 305
-L279: "Casparian strip": is Mn blocked by this band? Please see additional text on lines 311-312 and the new reference 48 which mentions the role of the Casparian strip in regulating trafficking of metal ions like iron, zinc and manganese.
Round 2
Reviewer 2 Report
Thanks for the updated manuscript.
I am very sorry, but I still feel that it is not adequate to show a textbook figure as concluding scheme for Mn transfer without referring to the effect of the bacterial inoculation (the topic of the manuscript). I suggest to either delete the figure, or present it as figure 1 and describe it in the introduction or include a (hypothetical) part on the effect of bacterial inoculation on Mn transfer.
Author Response
At the reviewer's request I have deleted Figure 7 from the manuscript.